# The role of depression in secondary HIV transmission among people who inject drugs in Vietnam: A mathematical modeling analysis

**Sara N. Levintow**[1]*, **Brian W. Pence**[1], **Teerada Sripaipan**[2], **Tran Viet Ha**[3], **Viet Anh Chu**[3], **Vu Minh Quan**[4], **Carl A. Latkin**[5], **Vivian F. Go**[2], **Kimberly A. Powers**[1]

**1** Department of Epidemiology, Gillings School of Global Public Health, University of North Carolina, Chapel Hill, North Carolina, United States of America, **2** Department of Health Behavior, Gillings School of Global Public Health, University of North Carolina, Chapel Hill, North Carolina, United States of America, **3** UNC Project Vietnam, University of North Carolina, Hanoi, Vietnam, **4** Centers for Disease Control and Prevention, Atlanta, Georgia, United States of America, **5** Department of Health, Behavior and Society, Johns Hopkins Bloomberg School of Public Health, Baltimore, Maryland, United States of America

* levintow@email.unc.edu

## Abstract

### Background

Among people who inject drugs (PWID), depression burden is high and may interfere with HIV prevention efforts. Although depression is known to affect injecting behaviors and HIV treatment, its overall impact on HIV transmission has not been quantified. Using mathematical modeling, we sought to estimate secondary HIV transmissions and identify differences by depression among PWID.

### Methods

We analyzed longitudinal data from 455 PWID living with HIV in Vietnam during 2009–2013. Using a Bernoulli process model with individual-level viral load and behavioral data from baseline and 6-month follow-up visits, we estimated secondary HIV transmission events from participants to their potentially susceptible injecting partners. To evaluate differences by depression, we compared modeled transmissions per 1,000 PWID across depressive symptom categories (severe, mild, or no symptoms) in the three months before each visit.

### Results

We estimated a median of 41.2 (2.5th, 97.5th percentiles: 33.2–49.2) secondary transmissions from all reported acts of sharing injection equipment with 833 injecting partners in the three months before baseline. Nearly half (41%) of modeled transmissions arose from fewer than 5% of participants in that period. Modeled transmissions per 1,000 PWID in that period were highest for severe depressive symptoms (100.4, 80.6–120.2) vs. mild (87.0, 68.2–109.4) or no symptoms (78.9, 63.4–94.1). Transmission estimates fell to near-zero at the 6-month visit.

**Data Availability Statement:** All relevant data are within the paper and its Supporting information files.

**Funding:** This study and the parent trial were funded by the National Institute on Drug Abuse (R36 DA045569 awarded to S.N.L.; R01 DA022962 awarded to V.F.G.; https://nida.nih.gov/). Doctoral training support for S.N.L. was provided by the National Institute of Allergy and Infectious Diseases (T32 AI070114; https://www.niaid.nih.gov/) and ViiV Healthcare (pre-doctoral fellowship; https://viivhealthcare.com/). The funders had no role in study design, data collection and analysis, decision to publish, or preparation of the manuscript.

**Competing interests:** The authors have declared that no competing interests exist.

## Conclusions

Secondary transmissions were predicted to increase with depression severity, although most arose from a small number of participants. Our findings suggest that effective depression interventions could have the important added benefit of reducing HIV transmission among PWID.

## Introduction

Sharing drug preparation and injection equipment is one of the most efficient means of HIV transmission, enabling bodily fluids from a person with HIV to directly enter an injecting partner's bloodstream [1, 2]. Globally, the estimated prevalence of HIV among people who inject drugs (PWID) is 18% [3], and injection drug use accounts for nearly one-third of incident HIV infections outside of sub-Saharan Africa [4, 5]. Unfortunately, coverage of harm reduction services for most people who inject drugs (PWID) remains inadequate [6–8], and barriers to HIV care engagement and antiretroviral therapy (ART) use can inhibit viral suppression, resulting in onward HIV transmission to injecting partners [6, 7, 9]. Equipment sharing in the absence of viral suppression continues to sustain the HIV epidemic among PWID, particularly in Asia and eastern Europe [10, 11].

Depression may amplify HIV transmission among PWID [12, 13]. Depression prevalence estimates among PWID range from 29% to 75% [14–18], considerably higher than pooled prevalence estimates of 13%-24% in other populations with HIV [19] and global estimates that 4.4% of the world's population experiences depression [20]. Depression has been linked to both increases in sharing behaviors [12, 21–24] and decreases in ART initiation and HIV viral suppression [13, 25–28]. Despite evidence supporting these links, the role of depression in HIV transmission among PWID has not been quantified. Prior research has focused on individual components of transmission risk (i.e., sharing behaviors *or* viral load) but has not investigated the overall contribution of depression to HIV transmission through the combination of these behavioral and biological pathways.

In this study, we use mathematical modeling to: 1) estimate secondary HIV transmission events from PWID living with HIV to their potentially susceptible injecting partners, and 2) examine differences in transmissions by depression status. We base input parameter values on behavioral and viral load data from PWID in Vietnam, where injection drug use drives the HIV epidemic [29–31]. Given continued HIV incidence among PWID [9, 10] and the plausible role of depression, it is critical to better understand how depression may drive onward transmission and the extent to which successful depression treatment could avert future infections.

## Methods

### Study population

We modeled parenteral HIV transmissions arising from 455 adult, male PWID with HIV enrolling in a randomized controlled trial of an HIV prevention intervention in Thai Nguyen Province, Vietnam between 2009 and 2013. Participants were recruited via snowball sampling from the 32 sub-districts of Thai Nguyen with the most PWID. HIV testing was performed, and participants diagnosed with HIV were referred to ART clinics and enrolled in the trial. The intervention included structural- and individual-level components. The structural

intervention aimed to reduce community stigma toward HIV infection and injection drug use, and the individual intervention provided support to participants in coping with HIV and reducing condomless sex and sharing of injection drug use equipment. As previously reported [32], no differences in injecting or sexual behaviors by intervention arm were observed over 24 months in the trial.

## Measures

Participants responded to a questionnaire in face-to-face interviews at baseline and follow-up visits at 6, 12, 18, and 24 months. At each interview, participants were asked to name up to ten partners with whom they had injected drugs (with or without shared equipment) in the prior three months, and for each partner, to report the frequency in those three months of sharing: 1) the same needle or syringe, 2) drug solution, and 3) the same ampoule of distilled water or novocain (a short-acting local anesthetic used to dissolve heroin). Participants were asked if each partner had HIV, with response options being "Yes", "No", or "Don't know." The questionnaire also assessed depressive symptoms in the past week with the Center for Epidemiologic Studies Depression Scale (CES-D), a measurement tool that has been widely used in populations with drug use [15, 33–35] and shown to have strong reliability and validity in a similar study population in Vietnam [36]. Although not a clinical diagnostic, the CES-D has good sensitivity and specificity for detecting depression in the general population and among patients with comorbid chronic illness, such as HIV [18, 36–38]. Consistent with this past work, we defined severe symptoms as CES-D scores $\geq 23$, mild symptoms as scores 16–22, and no symptoms as scores $<16$.

In the trial, blood specimens were collected to confirm HIV infection and measure CD4 cell count. To obtain viral load data for this analysis, we performed HIV RNA testing on stored plasma specimens using COBAS® AmpliPrep/COBAS® TaqMan® HIV-1 Test v2.0 (Roche Diagnostics GmbH) with a lower limit of detection of 20 copies/mL. Due to insufficient volume in $>50\%$ of plasma samples from the 12-, 18-, and 24-month visits, we limited the scope of our analysis to data collected at the baseline and 6-month visits.

## Descriptive analysis

We performed descriptive analyses on viral load and behavioral data for the overall study population and by baseline depressive symptoms, including a summary of HIV susceptibility and exposure among injecting partners. "Potentially susceptible" partners were defined as those for whom the participant reported HIV-negative or unknown HIV status. Potentially susceptible partners were classified as "minimally exposed" if $\geq 1$ sharing act occurred with a participant who had suppressed viral load ($<400$ copies/ml), but no acts occurred with a participant who had unsuppressed viral load ($\geq 400$ copies/ml). Potentially susceptible partners with $\geq 1$ sharing act with a participant who was virologically unsuppressed were classified as "more exposed," given the much higher likelihood of transmission under conditions of unsuppressed vs. suppressed viral loads [39].

## Model equations

We developed a Bernoulli process model [40–45] to estimate the number of secondary HIV transmission events from each participant in two periods: the three months before baseline and the three months before the 6-month follow-up visit. Transmission estimates were based on viral load measurements taken at the end of a given period, reported numbers of injecting partners in the period, sharing acts within those partnerships over the period, and reported partner HIV status. With this model, we estimated the probability $P_{ij}$ that participant $i$ would

transmit HIV to named partner $j$ in a specified period, given the probability $\pi_j$ that the partner was already HIV-infected at the start of that period, the probability $\beta_i$ of HIV transmission in one sharing act with a susceptible partner, and the reported number of sharing acts $n_{ij}$ between participant and partner over the period:

$$P_{ij} = \left(1 - \pi_j\right)[1 - (1 - \beta_i)^{n_{ij}}] \tag{1}$$

We specified the per-act transmission probability $\beta_i$ as a function of viral load $v_i$, an average viral load set-point $v_0$, and the transmission probability $\beta_0$ corresponding to that set-point. In this expression, $\beta_i$ increases relative to $\beta_0$ at a rate $\alpha$ per $\log_{10}$-unit increase in viral load $v_i$ above $v_0$ [46]:

$$\beta_i = \alpha^{\log_{10}\left(\frac{v_i}{v_0}\right)}\beta_0 \tag{2}$$

We then estimated $\lambda_i$, the total modeled number of secondary transmissions for each participant $i$ across all partners $j$ in a given three-month period:

$$\lambda_i = \sum_j P_{ij} \tag{3}$$

Finally, we summed the $\lambda_i$ values across participants in each of the two periods for a total number of estimated transmissions in each window.

## Model parameters

The probability ($\pi_j$) of partner HIV infection at the start of a given period, along with the number of sharing acts ($n_{ij}$) with that partner over the period, were based on questionnaire reports at baseline and 6 months. We assigned $\pi_j$ a value of 0 if the partner was reportedly HIV-negative and 1 if a partner was reportedly HIV-positive. For partners whose HIV status was unknown, we specified $\pi_j$ as the estimated HIV prevalence (34%) among PWID in Thai Nguyen during the study period [30, 31, 47]. Participant viral load ($v_i$) was fully observed at baseline. For missing viral load due to insufficient samples at 6 months (31% of participants), we used multiple imputation by chained equations [48, 49] to impute viral suppression status and assign viral load in 50 imputed datasets. We used estimates in the scientific literature for the average viral load set-point ($v_0$ = 4.5 $\log_{10}$ copies/ml) [50], transmission probability per sharing act at that set-point ($\beta_0$ = 0.008) [51], and increases or decreases in infectiousness per $\log_{10}$ change in viral load relative to the set-point ($\alpha$ = 2.09) [46]. Further details about parameters are in S1 Table.

Prior modeling of HIV transmission via shared injecting materials [52–57] has relied on a single per-act probability value that was estimated without specification of (or differentiation by) type of sharing act [51]. Consistent with this work, we conducted one set of model analyses in which we included all types of sharing acts (needles or syringes, drug solutions, and ampoules) as potential transmission events, using the single value of $\beta_0$ for all types. Due to uncertainty about the extent to which the standard probability estimate reflects all acts vs. needle- or syringe-sharing specifically, we also conducted a set of analyses in which we included only needle- or syringe-sharing acts in the model.

## Modeling analysis

To incorporate uncertainty around model inputs into our model outputs, we ran the model 50 times for each of 50 imputed datasets (2,500 runs total). In each run, we drew values for $\pi_j$ (for partners with unknown HIV status), $\alpha$, and $\beta_0$ from the distributions specified by published

summary estimates and confidence limits (S1 Table). Participants with competing events between baseline and 6 months (12% died or incarcerated) or missed 6-month visits not due to a competing event (5%) were excluded from the analysis at 6 months.

To evaluate differences in modeled transmission events by depression, we stratified the population according to baseline depressive symptoms (severe, mild, or no symptoms), and in each time window, we compared the total numbers of transmissions, the number of transmissions per participant, and the probability of infection acquisition per partner with ≥1 sharing act. To produce estimates that could be interpreted beyond our study sample, we also calculated expected numbers of transmissions in each time window per 1,000 PWID living with HIV in a given depressive symptom category (severe, mild, no symptoms). In addition, because depressive symptoms were assessed at both baseline and 6 months, we conducted an analysis of 6-month transmission by 6-month depressive symptoms (instead of baseline symptoms).

To account for possible misreporting of partner HIV status, we performed a sensitivity analysis in which a random sample of 25% of partner HIV status reports in each model run were assumed to be inaccurate (i.e., status was reversed for 25% of partners reported to be HIV-positive or HIV-negative). We also performed a sensitivity analysis around the assumed relationship between viral load and the transmission probability, as the current estimate for $\alpha$ is based on heterosexual transmission and may not strictly hold in the context of injection drug use, where viral load increases above a certain minimum threshold may not affect transmission risk to the extent seen in heterosexual transmission. For this analysis, instead of deriving the probability $\beta_i$ from viral load $v_i$, we used the standard probability estimate $\beta_0$ [51] for all viral load values. All analyses were conducted using R Version 3.6.3 [58].

## Ethics

The trial and this analysis were approved by the ethical review committees at the Thai Nguyen Center for Preventive Medicine, the Johns Hopkins Bloomberg School of Public Health, and the University of North Carolina at Chapel Hill Gillings School of Global Public Health. The trial was registered at ClinicalTrials.gov (NCT01689545). Written informed consent was obtained from participants.

## Results

Among the 455 male participants with HIV at baseline, the median age was 35 years, almost half (47%) were married or cohabitating, and about two-thirds (69%) were employed full-time. Three-quarters (74%) became newly HIV diagnosed at enrollment, and 41% had a CD4 cell count <200 cells/μL. Most (73%) reported sharing injection drug use equipment with partners over the prior three months. Nearly half (44%) had severe depressive symptoms, 25% had mild symptoms, and 30% had no symptoms.

### Descriptive analyses

At baseline, participants had a median viral load of 4.3 $\log_{10}$ copies/mL and reported a total of 833 injecting partners (mean 1.8 per participant) in the prior three months (Table 1). Participants reported ≥1 act of sharing injection equipment with 578 of 833 partners. Among those partners with ≥1 sharing act, 45% were reported to have unknown HIV status, 28% were reported to be HIV-negative, and 27% were reported to already have HIV. On average, participants reported 22.3 sharing acts per partner (primarily sharing ampoules/solutions, rather than needles/syringes). At the 6-month visit, median viral load decreased to 3.6 $\log_{10}$ copies/mL, and the average number of injecting partners over months 3–6 dropped to 0.7 per

Table 1. Transmission-related characteristics of study population.

| | Overall | Baseline depressive symptoms | | |
| --- | --- | --- | --- | --- |
| | | Severe | Mild | None |
| | N = 455 | n = 202 | n = 115 | n = 138 |
| **Baseline characteristics** | | | | |
| Median (IQR) participant viral load | 4.3 (2.6, 4.9) | 4.5 (3.1, 5.0) | 4.1 (1.9, 4.7) | 4.4 (2.2, 4.9) |
| Number of injecting partners in last 3 months | 833 | 400 | 211 | 222 |
| Number with ≥1 sharing acts (% all partners) | 578 (69%) | 307 (77%) | 143 (68%) | 128 (58%) |
| HIV status (% partners with sharing) | | | | |
| Reported to be HIV-positive | 27% | 32% | 25% | 19% |
| Reported to be HIV-negative | 28% | 24% | 24% | 38% |
| Reported to be unknown | 45% | 44% | 50% | 43% |
| Mean (range) acts per partner with sharing | | | | |
| Sharing ampoules | 7.9 (0, 90) | 7.0 (0, 90) | 8.5 (0, 90) | 9.7 (0, 90) |
| Sharing drug solutions | 13.1 (0, 90) | 11.8 (0, 90) | 13.7 (0, 90) | 15.4 (0, 90) |
| Sharing needles or syringes | 1.3 (0, 30) | 1.4 (0, 30) | 1.8 (0, 30) | 0.6 (0, 30) |
| All acts | 22.3 (2, 188) | 20.2 (2, 182) | 23.9 (2, 180) | 25.6 (2, 188) |
| **6-month characteristics** | | | | |
| Number (%) for follow-up | | | | |
| Attended visit | 377 (83%) | 158 (78%) | 97 (84%) | 122 (88%) |
| Missed visit: competing event | 55 (12%) | 39 (19%) | 8 (7%) | 8 (6%) |
| Missed visit: no competing event | 23 (5%) | 5 (2%) | 10 (9%) | 8 (6%) |
| Median (IQR) participant viral load* | 3.6 (1.8, 4.5) | 3.8 (2.0, 4.6) | 3.4 (1.8, 4.3) | 3.7 (1.6, 4.5) |
| Number of injecting partners in last 3 months | 251 | 117 | 69 | 65 |
| Number with ≥1 sharing acts (% all partners) | 106 (42%) | 53 (45%) | 33 (48%) | 20 (31%) |
| HIV status (% partners with sharing) | | | | |
| Reported to be HIV-positive | 61% | 62% | 67% | 50% |
| Reported to be HIV-negative | 17% | 21% | 15% | 10% |
| Reported to be unknown | 22% | 17% | 18% | 40% |
| Mean (range) acts per partner with sharing | | | | |
| Sharing ampoules | 3.4 (0, 30) | 3.0 (0, 30) | 3.2 (0, 30) | 5.0 (0, 30) |
| Sharing drug solutions | 5.8 (0, 30) | 4.6 (0, 30) | 10.0 (0, 30) | 2.2 (0, 30) |
| Sharing needles or syringes | 0.9 (0, 12) | 1.2 (0, 8) | 0.4 (0, 8) | 1.1 (0, 12) |
| All acts | 10.2 (2, 72) | 8.8 (2, 45) | 13.6 (2, 60) | 8.3 (2, 72) |

* Among 377 participants who attended the 6-month visit, 139 had missing data on viral load due to insufficient sample volume.

participant. Among the 106 partners with ≥1 sharing act (out of 251 partners total), a higher percentage was reported to be HIV-positive (61%) and fewer sharing behaviors took place (mean 10.2 acts per partner). At baseline, 333 of 833 partners (40%) were classified as "potentially susceptible" and "more exposed" to HIV, decreasing to only 20 of 251 partners (8%) at 6 months (Fig 1).

At baseline and 6 months, there were few clear differences in viral load or sharing behaviors according to baseline depressive symptoms, except for numbers of injecting partners and partner HIV status (Table 1). Participants with severe symptoms reported approximately twice as many partners as participants with mild or no symptoms at both time points. At baseline, partners of participants with severe symptoms were more likely to have shared injection equipment (77%) compared to partners of participants with mild symptoms (68%) or no symptoms (58%); however, these partners were also more likely to be reported as already HIV-positive

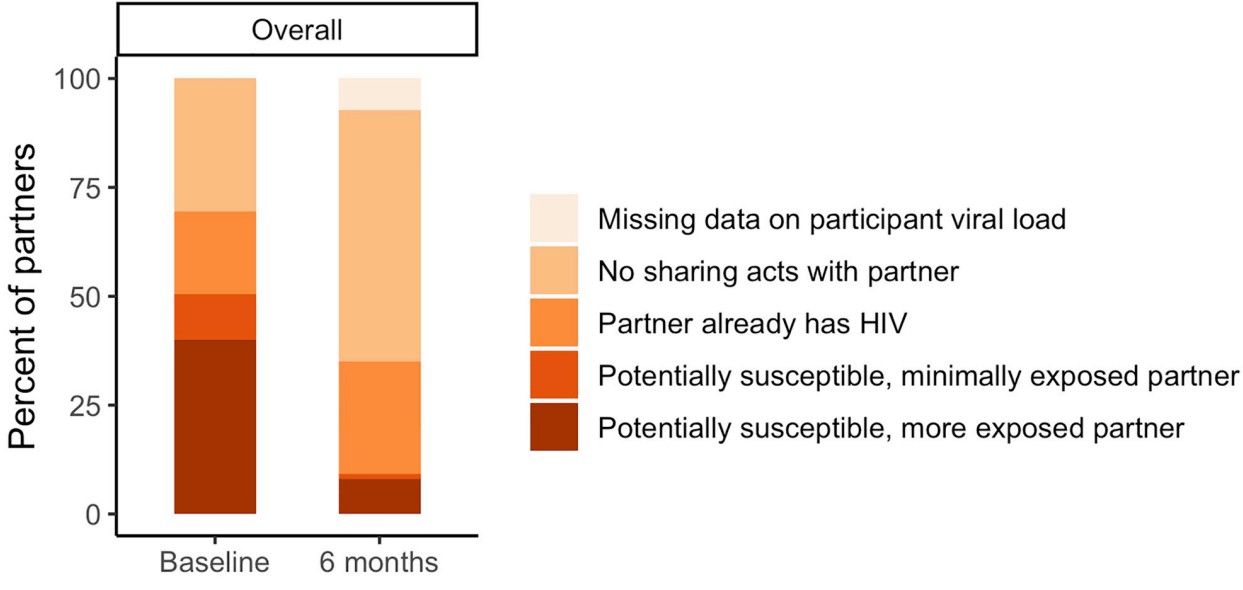

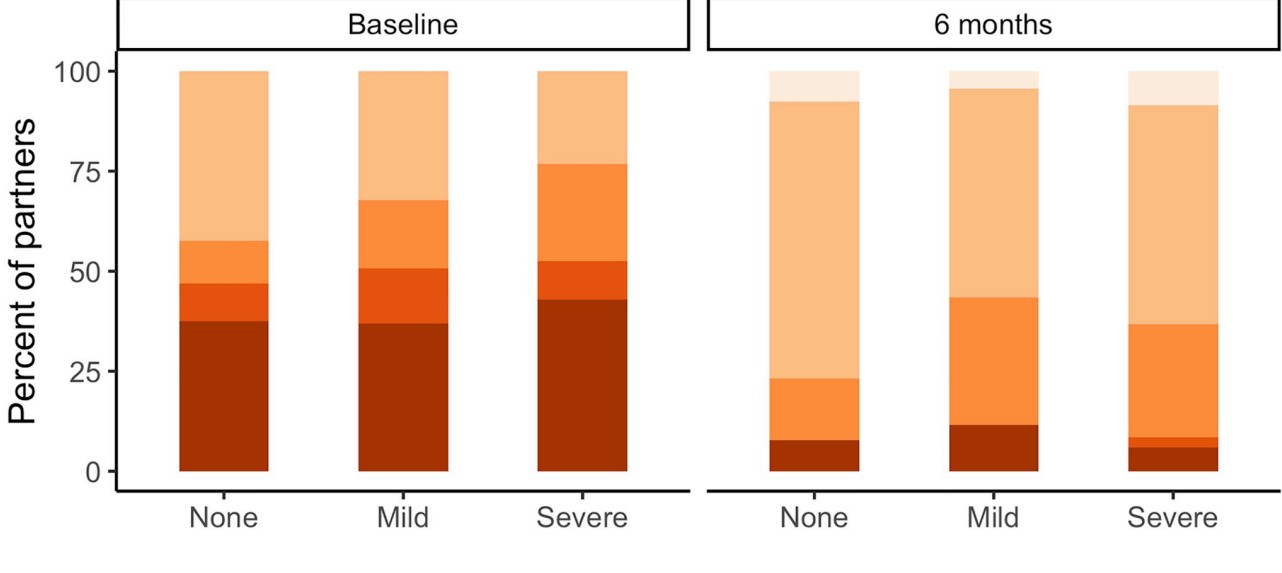

**Fig 1. Distribution of partner susceptibility and exposure to HIV, overall and by participant's baseline depressive symptoms.** Percentages are based on the 833 partners reported by 455 participants at the baseline visit and the 251 partners reported by 377 participants at the 6-month visit. "Potentially susceptible" refers to partners with reported HIV-negative or unknown HIV status. Among potentially susceptible partners, "minimally exposed" had ≥1 sharing act with a participant who had suppressed viral load (<400 copies/ml). "More exposed" partners participated in ≥1 sharing act with a participant who had unsuppressed viral load (≥400 copies/ml).

(32% for participants with severe symptoms vs. 25% and 19% for mild and no symptoms, respectively). Overall, 43% of partners of participants with severe symptoms were classified as potentially susceptible and more exposed to HIV (vs. 37% for mild or no symptoms) at baseline, compared to 6%, 12%, and 8% of partners reported at 6 months by participants with severe, mild, or symptoms, respectively (Fig 1).

**Table 2. Total modeled secondary transmission events in study population (N = 455) and expected number of transmission events per 1,000 PWID living with HIV, overall and in each depressive symptom category.**

| | Overall | Baseline depressive symptoms | | |
| --- | --- | --- | --- | --- |
| | | Severe | Mild | None |
| | N = 455 | n = 202 | n = 115 | n = 138 |
| **3 months prior to baseline** | | | | |
| All acts, total in study population | 41.2 (33.2–49.2) | 20.3 (16.3–24.3) | 10.0 (7.8–12.6) | 10.9 (8.8–13.0) |
| Needle-sharing only, total in population | 3.0 (2.2–3.8) | 2.2 (1.5–3.0) | 0.5 (0.4–0.7) | 0.3 (0.2–0.4) |
| All acts, number per 1,000 PWID | 90.6 (72.9–108.1) | 100.4 (80.6–120.2) | 87.0 (68.2–109.4) | 78.9 (63.4–94.1) |
| Needle-sharing only, per 1,000 PWID | 6.6 (4.9–8.4) | 10.7 (7.3–14.6) | 4.4 (3.1–6.5) | 2.2 (1.6–2.7) |
| **Follow-up months 3–6** | | | | |
| All acts, total in study population | 1.2 (0.8–1.7) | 0.8 (0.5–1.2) | 0.3 (0.2–0.5) | 0.1 (0.1–0.2) |
| Needle-sharing only, total in population | 0.1 (0.0–0.2) | 0.1 (0.1–0.2) | 0.0 (0.0–0.0) | 0.0 (0.0–0.0) |
| All acts, number per 1,000 PWID | 2.6 (1.7–3.8) | 4.9 (2.9–7.4) | 3.3 (2.1–5.5) | 0.6 (0.4–1.5) |
| Needle-sharing only, per 1,000 PWID | 0.3 (0.1–0.4) | 0.8 (0.3, 1.2) | 0.0 (0.0–0.0) | 0.0 (0.0–0.0) |

Estimates are reported as the median (2.5th-97.5th percentiles) from 2,500 model runs and are estimated separately by type of sharing act (all sharing acts vs. only needle-/syringe-sharing).

## Modeling analyses

Across the 2,500 model runs, we estimated a median of 41.2 (2.5th-97.5th percentiles: 33.2–49.2) secondary transmissions from all sharing acts with 833 injecting partners in the three months prior to baseline (Table 2). This estimate fell to 3.0 (2.2–3.8) transmissions in analyses where only needle-/syringe-sharing acts were modeled. For 251 partners reported at 6 months, we estimated a median of 1.2 (0.8–1.7) transmissions in the prior three months based on all acts and 0.1 (0.0–0.2) based on needle-/syringe-sharing only.

The highest numbers of transmissions were predicted for participants with severe depressive symptoms. Of an estimated 41.2 transmissions from all acts in the three months prior to baseline, 20.3 (16.3–24.3) were predicted to arise from participants with severe symptoms, 10.0 (7.8–12.6) from participants with mild symptoms, and 10.9 (8.8–13.0) from participants with no symptoms (Table 2). Transmissions per 1,000 PWID living with HIV in the baseline period were highest for severe symptoms (100.4, 80.6–120.2), compared to mild symptoms (87.0, 68.2–109.4) and no symptoms (78.9, 63.4–94.1) (Table 2, Fig 2). When modeled transmissions were restricted to needle-/syringe-sharing, 2.2 (1.5, 3.0) of 3.0 transmissions at baseline were predicted to arise from participants with severe symptoms (vs. <1 for participants with mild or no symptoms). Similarly, transmissions per 1,000 PWID remained highest for severe symptoms (10.7, 7.3–14.6), compared to mild symptoms (4.4, 3.1–6.5) and no symptoms (2.2, 1.6–2.7).

In follow-up months 3–6, the 1.2 transmissions modeled for all acts primarily arose from participants with severe symptoms (0.8, 0.5–1.2) (Table 2). Transmissions per 1,000 PWID in months 3–6 were lower than baseline estimates but showed a similar relationship with depressive symptoms: 4.9 (2.9–7.4) for severe symptoms, 3.3 (2.1–5.5) for mild symptoms, and 0.6 (0.4–1.5) for no symptoms (Table 2, Fig 3). Estimates further decreased when only needle-/syringe-sharing acts were modeled; non-zero transmissions were predicted only for participants with severe symptoms. Results were similar when the 6-month depressive symptom measure was used (S1 Fig).

When we accounted for possible misreporting of partner HIV status, there was greater uncertainty in transmission estimates but differences by depression were similar

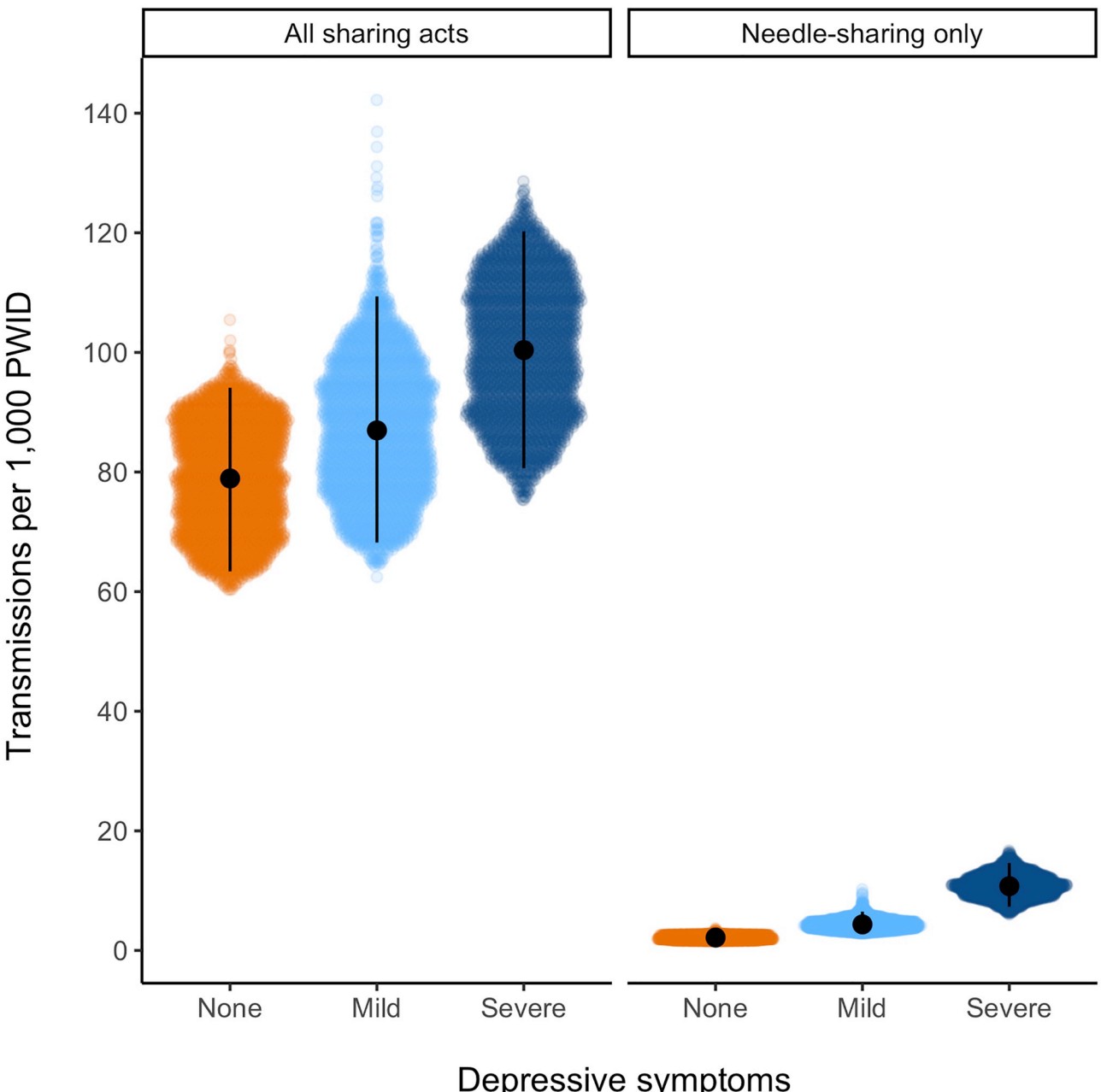

**Fig 2. Modeled secondary transmission events in the three months prior to baseline per 1,000 PWID living with HIV in each depressive symptom category.** Estimates from 2,500 model runs are plotted by type of sharing act (all sharing acts vs. only needle-/syringe-sharing). Colored regions show all 2,500 estimates, the black point is the median, and the vertical line extends from the 2.5th to 97.5th percentiles.

(S2A and S2B Fig). When we assumed a constant transmission probability across viral load (S3A and S3B Fig), more transmissions at baseline were predicted for participants with mild vs. severe symptoms. This divergence from the main results likely relates to slightly higher viral loads among participants with severe (vs. mild) symptoms, resulting in higher transmission probabilities for severe (vs. mild) symptoms in the main analysis but not this sensitivity analysis.

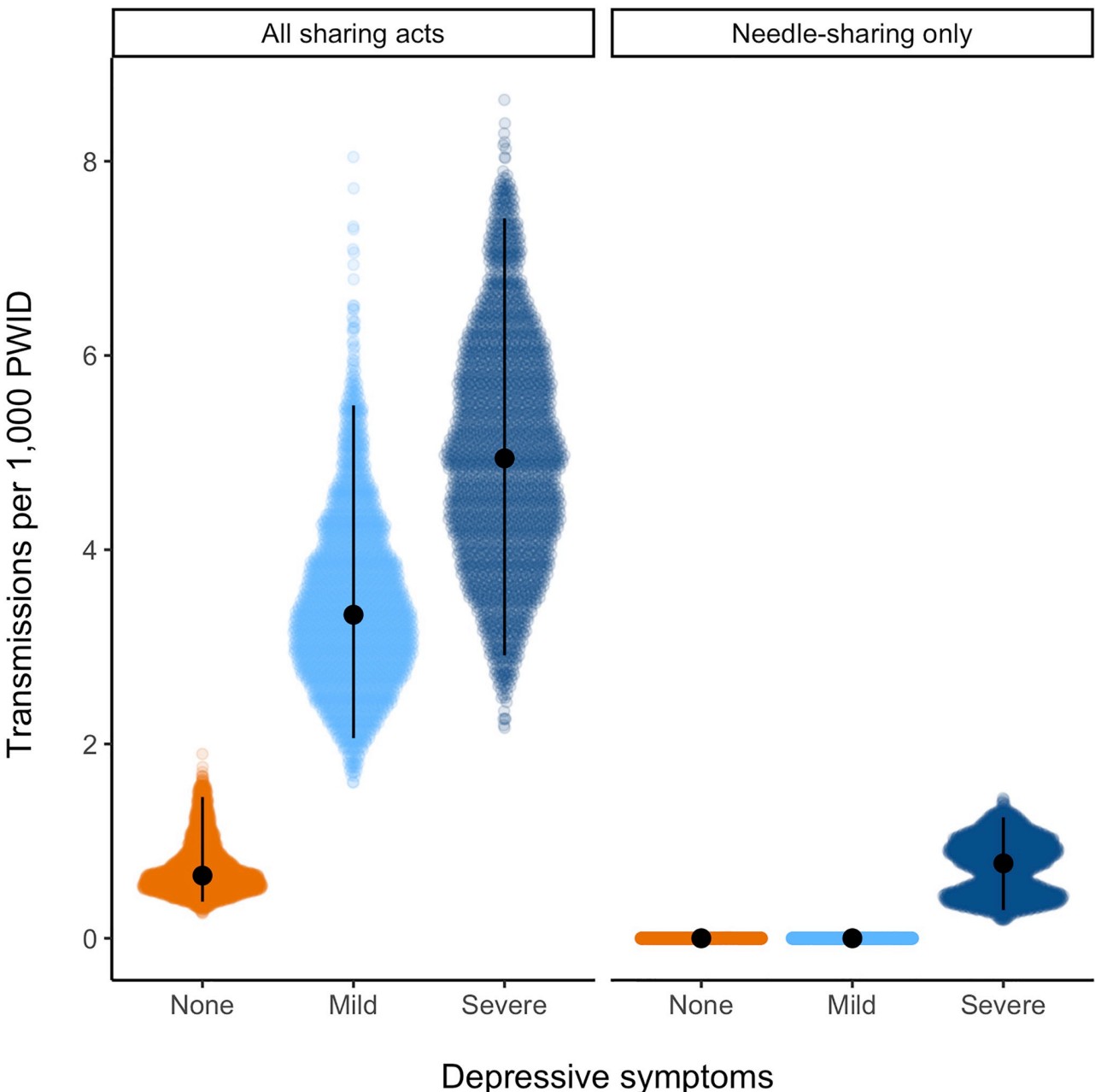

**Fig 3. Modeled secondary transmission events in follow-up months 3–6 per 1,000 PWID living with HIV in each depressive symptom category.**
Estimates from 2,500 model runs are plotted by type of sharing act (all sharing acts vs. only needle-/syringe-sharing). Colored regions show all 2,500 estimates, the black point is the median, and the vertical line extends from the 2.5th to 97.5th percentiles.

Nearly half of modeled transmissions in the three months prior to baseline (17.1 of 41.2, 41%) arose from just 20 of 455 participants; in months 3–6, only 25 participants accounted for all modeled transmissions. Given that most participants had zero predicted transmissions (due to no sharing or ≥1 acts with only HIV-positive partners), the mean probability of infection per partner and mean transmissions per participant were heavily weighted toward zero regardless of depressive symptoms (S4 and S5 Figs). When modeling transmission using all acts at baseline, the mean probability of infection per partner (given ≥1 act) was 0.07–0.09 across

depressive symptom categories, falling closer to zero when only needle-/syringe-sharing acts were modeled and across sharing acts at 6 months. However, there were outliers with higher mean probabilities, particularly for partners of participants with severe symptoms, and to some extent, partners of participants with mild symptoms. The mean transmissions per participant ranged from 0–3 across time points, types of sharing acts, and depressive symptoms. Among the 20 participants with ≥0.5 transmissions at baseline (resulting in nearly half of transmissions), 55% had severe symptoms (vs. 44% overall prevalence of severe symptoms at baseline). Among the 25 participants accounting for all modeled transmissions at 6 months, 52% had severe symptoms (vs. 42% overall prevalence of severe symptoms at 6 months).

## Discussion

In this study of PWID living with HIV in Vietnam, we used mathematical modeling to estimate secondary HIV transmission events from PWID to their potentially susceptible injecting partners in two short windows. A high number of secondary transmissions was predicted for the three months prior to baseline when transmission was modeled for all sharing behaviors (i.e., sharing needles or syringes, drug solutions, and ampoules). This estimate dramatically decreased when only needle-/syringe-sharing behaviors were included. These disparate findings demonstrate the sensitivity of model estimates to the assumed infectivity of different sharing behaviors, a phenomenon that has not been routinely considered in prior modeling studies of injection-related HIV transmission. Due to substantial declines in viremia and sharing behaviors after baseline (i.e., after HIV diagnosis and referral to ART clinics), we estimated that very little secondary transmission occurred during follow-up months 3–6, underscoring the importance of diagnosis and treatment for HIV prevention in PWID populations.

Prior research has linked depression to increases in sharing behaviors [12, 21–24] and decreases in ART initiation and viral suppression [13, 25–28], and in the current analysis, we predicted more secondary transmissions for PWID with depression. Across our main analyses, estimated transmissions in each time period were highest for participants with severe symptoms. However, there was substantial variability in the magnitude of differences across model runs, time points assessed, and specific behaviors modeled. The majority of modeled transmissions arose from a small number of participants, resulting in group averages of the probability of infection per partner and transmissions per participant being near-zero regardless of depressive symptoms. Given the multi-faceted nature of transmission, it is plausible that depression has a combination of synergistic and antagonistic effects, with the contribution of each varying across individuals.

Future mathematical models could investigate the extent to which successful depression treatment might reduce continued HIV incidence, assuming that decreasing the severity of symptoms to mild or none through treatment would translate to the lower transmissions predicted for those groups in our study. This work could incorporate our study's estimates of viral load and sharing behaviors by depressive symptoms and the corresponding differences in transmissions to injecting partners. Depression treatment at varying coverage and efficacy levels could be modeled alongside other prevention interventions (ART, needle exchanges, methadone maintenance) to understand the optimal combination of these approaches in decreasing HIV incidence. Future models of injection-related HIV transmission should also examine uncertainties in the infectivity of different sharing behaviors and the relationship between viral load and transmission probability in the context of injection drug use. Assumptions along these dimensions substantially influenced our transmission estimates, underscoring the importance of understanding these understudied phenomena.

Our mathematical modeling approach enabled the integration of behavioral and biological mechanisms to estimate HIV transmissions according to depression status, but we did not conduct a formal causal analysis with consideration of potential confounders or effect measure modifiers. We also expect that our model predictions underestimate all HIV transmissions arising from this population during the study period. Social desirability bias may have led to underestimates of sensitive sharing behaviors, and missing data precluded estimates for participants who did not attend the 6-month visit. Further, we focused on injection-related HIV transmission to injecting partners and excluded sexual transmission, as condomless sex was rarely reported with injecting partners in our study population. We also note that our model assumes that viral load measures taken at baseline and 6 months applied to the prior three months, which might have been especially likely to underestimate transmission in the second interval, when viral loads may have been decreasing over time due to diagnosis and treatment. Additional underestimation of transmission could have resulted from our assumption that partners reported to be HIV-positive at the end of an interval were HIV-positive at the beginning of the interval. Misreporting of partner HIV status and sharing acts may have been differential by depression status, as those with low levels of reported depressive symptoms may be less likely to report socially undesirable behaviors [59]. As a further limitation, we note that relationships between depressive symptoms and transmission may differ in the current epidemic era and in populations outside our study setting.

Despite these limitations, our modeling analyses suggest a skewed distribution of transmission by depression status, with secondary transmissions predicted to increase with symptom severity and a disproportionate burden of severe depressive symptoms among those participants from whom transmissions were predicted to arise. Our findings reflect a complex relationship of depression with HIV transmission and suggest that depression interventions have the potential to decrease secondary HIV transmissions, in addition to providing crucial mental health benefits among PWID.

## Supporting information

**S1 Table. Bernoulli model input parameters.**
(DOCX)

**S1 Fig. Modeled secondary transmission events in follow-up months 3–6 per 1,000 PWID living with HIV, based on depressive symptoms assessed at 6 months instead of baseline.** Estimates from 2,500 model runs are plotted by type of sharing act (all sharing acts vs. only needle-/syringe-sharing). Colored regions show all 2,500 estimates, the black point is the median, and the vertical line extends from the 2.5th to 97.5th percentiles.
(DOCX)

**S2 Fig.** A. Sensitivity analysis accounting for possible misreporting of partner HIV status (baseline transmission by baseline depression). B. Sensitivity analysis accounting for possible misreporting of partner HIV status (transmission in months 3–6 by baseline depression).
(ZIP)

**S3 Fig.** A. Sensitivity analysis allowing for constant transmission probability (baseline transmission by baseline depression). B. Sensitivity analysis allowing for constant transmission probability (transmission in months 3–6 by baseline depression).
(ZIP)

**S4 Fig. Mean probability of infection acquisition in prior 3 months per injecting partner with ≥1 sharing act reported at baseline and 6 months, by participant depressive**

**symptoms.** Each colored point shows one partner's mean probability across 2,500 model runs; dashed lines show the mean of means (across all partners and model runs). Mean probabilities are reported separately by the types of sharing acts included in the model (all sharing acts vs. only needle-/syringe-sharing). A) Baseline probabilities by baseline depression. B) 6-month probabilities by baseline depression. C) 6-month probabilities by 6-month depression.
(DOCX)

**S5 Fig. Mean numbers of secondary transmission events in prior 3 months per participant, by timepoint and depressive symptoms.** Each colored point shows one participant's mean transmissions across 2,500 model runs; dashed lines show the mean of means (across all participants and model runs). Mean transmissions are reported separately by the types of sharing acts included in the model (all sharing acts vs. only needle-/syringe-sharing). A) Baseline transmissions by baseline depression. B) 6-month transmission by baseline depression. C) 6-month transmission by 6-month depression.
(DOCX)

**S1 Dataset.**
(CSV)

## Acknowledgments

We thank Dr. Steven R. Meshnick and Dr. Stephen R. Cole for helpful discussions on the conception and design of the study.

## Author Contributions

**Conceptualization:** Sara N. Levintow, Brian W. Pence, Teerada Sripaipan, Tran Viet Ha, Viet Anh Chu, Vu Minh Quan, Carl A. Latkin, Vivian F. Go, Kimberly A. Powers.

**Formal analysis:** Sara N. Levintow.

**Funding acquisition:** Sara N. Levintow, Brian W. Pence, Carl A. Latkin, Vivian F. Go, Kimberly A. Powers.

**Investigation:** Sara N. Levintow, Brian W. Pence, Teerada Sripaipan, Tran Viet Ha, Viet Anh Chu, Vu Minh Quan, Carl A. Latkin, Vivian F. Go, Kimberly A. Powers.

**Methodology:** Sara N. Levintow, Brian W. Pence, Carl A. Latkin, Vivian F. Go, Kimberly A. Powers.

**Project administration:** Teerada Sripaipan, Tran Viet Ha, Viet Anh Chu, Vu Minh Quan.

**Supervision:** Brian W. Pence, Carl A. Latkin, Vivian F. Go, Kimberly A. Powers.

**Writing – original draft:** Sara N. Levintow.

**Writing – review & editing:** Sara N. Levintow, Brian W. Pence, Teerada Sripaipan, Tran Viet Ha, Viet Anh Chu, Vu Minh Quan, Carl A. Latkin, Vivian F. Go, Kimberly A. Powers.

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
