## [Decision Letter · Decision Letter 0]

19 Jul 2022

PONE-D-22-09164The role of depression in HIV transmission among people who inject drugs in Vietnam: A mathematical modeling analysisPLOS ONE

Dear Dr. Levintow,

Thank you for submitting your manuscript to PLOS ONE. After careful consideration, we feel that it has merit but does not fully meet PLOS ONE’s publication criteria as it currently stands. Therefore, we invite you to submit a revised version of the manuscript that addresses the points raised during the review process.

Please revise the manuscript based on the reviewers' comments.

We look forward to receiving your revised manuscript.

Kind regards,

Hamid Sharifi

Academic Editor

PLOS ONE

Journal Requirements:

“This study and the parent trial were funded by NIDA (R36 DA045569, R01 DA022962). Doctoral training support for S. N. Levintow was also provided by NIAID (T32 AI070114) and ViiV Healthcare (pre-doctoral fellowship). The funders had no role in study design, data collection and analysis, decision to publish, or preparation of the manuscript.”

“This study and the parent trial were funded by the National Institute on Drug Abuse (R36 DA045569 awarded to S.N.L.; R01 DA022962 awarded to V.F.G.; https://nida.nih.gov/). Doctoral training support for S.N.L. was provided by the National Institute of Allergy and Infectious Diseases (T32 AI070114; https://www.niaid.nih.gov/) and ViiV Healthcare (pre-doctoral fellowship; https://viivhealthcare.com/). The funders had no role in study design, data collection and analysis, decision to publish, or preparation of the manuscript.”

Additional Editor Comments:

Major Revision

Reviewers' comments:

Reviewer's Responses to Questions

**Comments to the Author**

1. Is the manuscript technically sound, and do the data support the conclusions?

Reviewer #1: Yes

Reviewer #2: Yes

Reviewer #3: Partly

2. Has the statistical analysis been performed appropriately and rigorously? 

Reviewer #1: Yes

Reviewer #2: Yes

Reviewer #3: Yes

3. Have the authors made all data underlying the findings in their manuscript fully available?

Reviewer #1: Yes

Reviewer #2: Yes

Reviewer #3: Yes

4. Is the manuscript presented in an intelligible fashion and written in standard English?

Reviewer #1: Yes

Reviewer #2: Yes

Reviewer #3: Yes

5. Review Comments to the Author

Reviewer #1: Date: June 18, 2022

Study title

The role of depression in HIV transmission among people who inject drugs in Vietnam: A mathematical modeling analysis

General comments

This is one of the relevant literatures that presented the mathematical modelling of Bernoulli process to describe and explain the depression and secondary HIV transmission. The manuscript is well structured, narrated and put in logical order, the flow of which is easy to go through to the end. The introduction section has clearly put the gray area or the problem statement. The result section is detailed, with relevant finding addressing the objective of the study. The authors have summarized the finding, showed the relevance of the finding to HIV prevention and control policy suggestions. The persisted gray area and the study limitations were also well explained and justified for the future investigation. The

Specific comments

The authors might consider the following items to further improve the manuscript

1. It seems that the mainstay of HIV transmission in Vietnam is among people who inject drugs. However, the main ways of HIV transmission could be presented in introduction/method section objectively, so that the reader could easily understand which section of population are at risk for HIV, and then the significant of the modelling could be justified.

2. The key modelling finding, and discussion was about depression and secondary HIV transmission which might be reflected in the title and objective of the study.

3. There seem old references (older than 2005). The authors are advised to look for recent references and replace with the recent relevant evidence.

Reviewer #2: This is an interesting article with very important implications. It is important to evaluate the psychosocial factors influencing behavior and this was very clear as to the conclusions and implications. I am not sophisticated to judge the modeling but will assume it was done properly.

Reviewer #3: This paper is on an important subject that correlates PWID, depression and HIV infection risk. The association of PWID and the risk for secondary transmission of HIV seem adequately addressed by modelling. However, the role of depression as a variable was not thoroughly investigated to answer the following questions:

1. What is the prevalence of depression in PWID compared with the other drug users and general population?

2. Were robust measurement parameters used to ascertain the depression status of the recruited participants?

3. Were confounders/effect modifiers considered in associating depression among PWID with the risk of secondary HIV transmission

Following these responses, bring out a clear take home message from the study, that address how the limitations of the model affect the conclusions

6. PLOS authors have the option to publish the peer review history of their article (what does this mean?). If published, this will include your full peer review and any attached files.

Reviewer #1: No

Reviewer #2: **Yes: **Richard Elion

Reviewer #3: **Yes: **dumiwaboka@gmail.com

---

## [Author Response · Author response to Decision Letter 0]

23 Aug 2022

Response to Reviewers

Reviewer 1 

This is one of the relevant literatures that presented the mathematical modelling of Bernoulli process to describe and explain the depression and secondary HIV transmission. The manuscript is well structured, narrated and put in logical order, the flow of which is easy to go through to the end. The introduction section has clearly put the gray area or the problem statement. The result section is detailed, with relevant finding addressing the objective of the study. The authors have summarized the finding, showed the relevance of the finding to HIV prevention and control policy suggestions. The persisted gray area and the study limitations were also well explained and justified for the future investigation. The specific comments the authors might consider the following items to further improve the manuscript.

1. It seems that the mainstay of HIV transmission in Vietnam is among people who inject drugs. However, the main ways of HIV transmission could be presented in introduction/method section objectively, so that the reader could easily understand which section of population are at risk for HIV, and then the significant of the modelling could be justified.

Response: We appreciate this suggestion from the reviewer and have revised the first paragraph of the Introduction to provide greater context for the modes of HIV transmission. This text now states: “Sharing drug preparation and injection equipment is one of the most efficient means of HIV transmission, enabling bodily fluids from a person with HIV to directly enter an injecting partner’s bloodstream (1,2). Globally, the estimated prevalence of HIV among people who inject drugs (PWID) is 18% (3), and injection drug use accounts for nearly one-third of incident HIV infections outside of sub-Saharan Africa (4,5).”

2. The key modelling finding, and discussion was about depression and secondary HIV transmission which might be reflected in the title and objective of the study.

Response: We updated the title to specify “secondary” HIV transmission, and we revised the objective statement in the abstract: “Using mathematical modeling, we sought to estimate secondary HIV transmissions and identify differences by depression among PWID.” We also clarified this objective in the first sentence of the last paragraph of the Introduction: “In this study, we use mathematical modeling to: 1) estimate secondary HIV transmission events from PWID living with HIV to their potentially susceptible injecting partners, and 2) examine differences in transmissions by depression status.”

3. There seem old references (older than 2005). The authors are advised to look for recent references and replace with the recent relevant evidence.

Response: We closely reviewed the references but found that the majority represent recent evidence on the HIV epidemic among PWID from the last 5-10 years. Only 4 of 60 references are older than 2005, and all 4 correspond to foundational work for the development of the methods used in this study. Radloff 1977 evaluated the psychometric properties of the CES-D for depression screening, and we include citations to more recent studies validating and applying the CES-D in Vietnamese populations (Thai et al. 2016, Huynh et al. 2017). Pinkerton & Abramson 1993 and Rottingen et al. 2002 developed the Bernoulli modeling methodology, and we also cite more recent studies using these models (Pearson et al. 2007, Fox et al. 2011, Adams et al. 2013, Kroon et al. 2017). Rubin 1987 developed the multiple imputation methodology, and we include a citation to more recent work programming these methods into statistical software (van Buuren & Groothuis-Oudshoorn 2011). 

Reviewer 2 

This is an interesting article with very important implications. It is important to evaluate the psychosocial factors influencing behavior and this was very clear as to the conclusions and implications. I am not sophisticated to judge the modeling but will assume it was done properly.

 Response: We thank the reviewer for positive feedback on our manuscript.

Reviewer 3 

This paper is on an important subject that correlates PWID, depression and HIV infection risk. The association of PWID and the risk for secondary transmission of HIV seem adequately addressed by modelling. However, the role of depression as a variable was not thoroughly investigated to answer the following questions:

1. What is the prevalence of depression in PWID compared with the other drug users and general population?

Response: We added greater background to the second paragraph of the Introduction on the prevalence of depression among PWID and in other populations. This text reads: “Depression prevalence estimates among PWID range from 29% to 75% (14–18), considerably higher than pooled prevalence estimates of 13%-24% in other populations with HIV (19) and global estimates that 4.4% of the world’s population experiences depression (20).” 

2. Were robust measurement parameters used to ascertain the depression status of the recruited participants?

Response: We expanded on the description of the psychometric properties of Center for Epidemiologic Studies Depression Scale (CES-D) in the Measures subsection of the Methods: “The questionnaire also assessed depressive symptoms in the past week with the Center for Epidemiologic Studies Depression Scale (CES-D), a measurement tool that has been widely used in populations with drug use (15,33–35) and shown to have strong reliability and validity in a similar study population in Vietnam (36). Although not a clinical diagnostic, the CES-D has good sensitivity and specificity for detecting depression in the general population and among patients with comorbid chronic illness, such as HIV (18,36–38).”

3. Were confounders/effect modifiers considered in associating depression among PWID with the risk of secondary HIV transmission?

Response: We appreciate the reviewer bringing up these methodological considerations, which we have now described as a limitation of our study in the Discussion: “Our mathematical modeling approach enabled the integration of behavioral and biological mechanisms to estimate HIV transmissions according to depression status, but we did not conduct a formal causal analysis with consideration of potential confounders or effect measure modifiers.” 

Given this study limitation, we also toned down some of the language in the Discussion paragraph on future research directions. Modeled interventions on depression would be hypothetical and could potentially reduce expected transmissions under certain assumptions, now described in the text: “Future mathematical models could investigate the extent to which successful depression treatment might reduce continued HIV incidence, assuming that decreasing the severity of symptoms to mild or none through treatment would translate to the lower transmissions predicted for those groups in our study.”

---

## [Decision Letter · Decision Letter 1]

27 Sep 2022

The role of depression in secondary HIV transmission among people who inject drugs in Vietnam: A mathematical modeling analysis

PONE-D-22-09164R1

Dear Dr. Levintow

We’re pleased to inform you that your manuscript has been judged scientifically suitable for publication and will be formally accepted for publication once it meets all outstanding technical requirements.

Kind regards,

Hamid Sharifi

Academic Editor

PLOS ONE

Additional Editor Comments (optional):

Reviewers' comments:

Reviewer's Responses to Questions

**Comments to the Author**

1. If the authors have adequately addressed your comments raised in a previous round of review and you feel that this manuscript is now acceptable for publication, you may indicate that here to bypass the “Comments to the Author” section, enter your conflict of interest statement in the “Confidential to Editor” section, and submit your "Accept" recommendation.

Reviewer #2: All comments have been addressed

2. Is the manuscript technically sound, and do the data support the conclusions?

Reviewer #2: Yes

3. Has the statistical analysis been performed appropriately and rigorously? 

Reviewer #2: Yes

4. Have the authors made all data underlying the findings in their manuscript fully available?

Reviewer #2: Yes

5. Is the manuscript presented in an intelligible fashion and written in standard English?

Reviewer #2: Yes

6. Review Comments to the Author

Reviewer #2: excellent work, i have no further comments. This work raises an excellent point of the syndemic of psychological state and risk management.

7. PLOS authors have the option to publish the peer review history of their article (what does this mean?). If published, this will include your full peer review and any attached files.

Reviewer #2: No

---

## [Editor Report · Acceptance letter]

4 Oct 2022

PONE-D-22-09164R1 

The role of depression in secondary HIV transmission among people who inject drugs in Vietnam: A mathematical modeling analysis 

Dear Dr. Levintow:

I'm pleased to inform you that your manuscript has been deemed suitable for publication in PLOS ONE. Congratulations! Your manuscript is now with our production department. 

Kind regards, 

on behalf of

Dr. Hamid Sharifi 

Academic Editor

PLOS ONE